# Transperitoneal vs extraperitoneal radical cystectomy: A systematic review and meta-analysis

Kevin Leonardo [1], Hendy Mirza[2], Doddy Hami Seno[2], Nugroho Purnomo[2], Andika Afriansyah[2], Moammar Andar Roemare Siregar [2]*

1 Faculty of Medicine, Department of Urology, Universitas Indonesia–Cipto Mangunkusumo Hospital, Jakarta, Indonesia, 2 Faculty of Medicine, Department of Surgery, Division of Urology, Persahabatan General Hospital - Universitas Indonesia, Jakarta, Indonesia

* andar.siregar@gmail.com

## Abstract

### Background

One of the most complex surgeries including radical cystectomy (RC) has a high rate of morbidity. The standard approach for the muscle-invasive bladder is conventional transperitoneal radical cystectomy. However, the procedure is associated with significant morbidities like ileus, urinary leak, bleeding, and infection. The aim of this study is to compare the transperitoneal RC approach with the extraperitoneal RC approach in the treatment of bladder cancer patients. The outcomes of this study are Operative time, Estimated Blood Loss, Hospital Stay, Post-Operative Ileus, Infection, and Major Complication (Clavien-Dindo Grade 3–5).

### Methods

PubMed, Cochrane Library, and Science Direct were systematically searched for different publications related to the meta-analysis. Keywords used for searching were Radical Cystectomy AND Extraperitoneal AND Transperitoneal up until 31st August 2022. The studies were screened for our eligibility criteria. Demographic parameters, perioperative variables, and postoperative complications were recorded and analyzed. The Newcastle-Ottawa Scale was used to evaluate the risk of bias in each study. The Review Manager (RevMan) software version 5.4.1 was used for statistical analysis.

### Results

Eight studies (3 laparoscopic and 5 open methods) involving 1207 subjects (588 patients using the extraperitoneal approach and 619 using the transperitoneal approach) were included. The incidence of postoperative ileus is significantly lower after the extraperitoneal approach compared to the transperitoneal approach (p < 0.00001). The two techniques did not differ in operative time, estimated blood loss, duration of hospital stay, total infection, and major complication events.

**Data Availability Statement:** All relevant data are within the paper and its Supporting Information files.

**Funding:** The author(s) received no specific funding for this work.

## Conclusion

This meta-analysis shows that extraperitoneal radical cystectomy benefits in terms of reduced postoperative ileus.

## Introduction

Bladder cancer is a carcinoma of the urothelial cells with an especially high incidence in male and elderly populations [1]. For decades, radical cystectomy (RC) with urinary diversion has been the standard treatment for non-metastatic muscle-invasive and high-risk non-muscle-invasive bladder cancer [2].

The transperitoneal approach is currently the most commonly used method, which involves transperitoneal antegrade mobilization of the bladder with blunt dissection [3]. Despite its popularity, this route has high complication rates (40–44%) [4]. The most common complication categories are gastrointestinal (29%), infectious (25%), and wound-related (15%), respectively. Ileus is the most common type of gastrointestinal complication (GC) [5].

A possible reason for this high rate is the contact of intestinal serosa with the de-peritonealised pelvic wall. This induces an inflammatory reaction that alongside postoperative adhesion bands, reduces bowel peristalsis, causes ileus, obstruction, distention, and increases pain [6]. Increased exposure of the intestines to the atmosphere and upward packing of bowel loops for clearing the operative field during the surgery contribute to the side effects. Hence, keeping the peritoneal continuity in this surgery has been reported as an important milestone in reducing postoperative complications [7].

In order to achieve peritoneal continuity, an ascending extraperitoneal technique with extra peritonealisation of the ileal-bladder has been developed in 1991. Since then, several studies have implied that extraperitoneal technique reduced ileus, re-operation rates, wound problems, ease of management of urinary leaks, and improved continence rates in neobladder patients [8]. However, there are concerns about the radicalness of this technique [6]. This study compares the perioperative parameters and also complications of the transperitoneal RC approach with the extraperitoneal RC approach in the treatment of bladder cancer patients.

## Methods

### Search strategy

The Preferred Reporting Items for Systematic Reviews and Meta-Analyses (PRISMA) checklist was used to perform this systematic review. The protocol is registered in the PROSPERO database (CRD42022360997). We conducted a comprehensive systematic review in three online databases (PubMed, Cochrane Library, and Science Direct) up until 31st August 2022. Radical Cystectomy AND Extraperitoneal AND Transperitoneal were used as main keywords, with several combinations and elaborations (S1 Table). Furthermore, we also reviewed the reference lists among several chosen articles and records from other search engines to identify additional relevant publications.

### Eligibility criteria

We included studies that provide information about the comparison of transperitoneal and extraperitoneal radical cystectomy. Inclusion of the paper included articles in English, full-text available, and published in the last 15 years. Exclusion criteria included review articles written

in languages other than English, conference abstracts, nonhuman research, and studies not evaluate the outcome measures. Any ambiguity or discrepancies were resolved by discussion among authors. The PRISMA flow diagram was used to guide the study selection process, and the authors approved the final list of selected papers to be included in this systematic review.

### Data extraction and outcome

Three independent reviewers conducted the study selection and inclusion process in two rounds (K. L, M. A. R. S, and A. A). Other reviewers (H. M, D. H. S, and N. P) were brought in to settle any disagreements and discrepancies. The key outcome measure was the choice of radical cystectomy approach, demographic parameters, perioperative variables, and postoperative complications. The Newcastle-Ottawa Scale was used to evaluate the risk of bias in each study. The risk of bias was evaluated for three variables: selection, comparability, and research outcome. Studies are considered to have a minimal risk of bias when they have a score of 7 or higher.

### Statistical analysis

Cochrane Collaboration Review Manager (RevMan version 5.4) was used to conduct the meta-analysis on the studies that included relevant outcome data. The type of meta-analysis model that we used was The Cochran-Mantel-Haenszel Method. The outcomes this study examined and assessed were operative time, estimated blood loss, length of hospital stay, ileus following surgery, overall infection event, and major complications. For operating time, estimated blood loss, duration of hospital stays, the standardized mean between-group difference, and respective 95% confidence intervals (CI) for each outcome were used to summarize the effects of the intervention for these continuous data. The latter outcomes, such as post-operative ileus, overall infection event, and major complication, were pooled, and the relative risk (RR) and their associated 95% CI were used to determine the value of these dichotomous data. P values less than 0.05 were regarded in every instance as statistically significant. $I^2$ was determined using RevMan version 5.4 in order to look into statistical heterogeneity. A percentage of more than 50% could be regarded as significant heterogeneity, so a random effects model was employed to estimate pooled effects; otherwise, a fixed effects model was used.

### Results

Databases searching identified a total of 250 articles (S1 Fig), and they were screened based on the inclusion and exclusion criteria included in the study selection. Of these, 24 articles passed the screening process and resulted in 18 articles for full-text assessment. Eight articles did not evaluate the outcome of interest, and we have to rule one article out due to combined intervention, using laparoscopic for the transperitoneal method and open for the extraperitoneal method (mix). Hence, we found eight appropriate studies included in this review (S2 Table).

Eight studies were successfully studied with a total sample of 1207 subjects (588 patients using the extraperitoneal approach and 619 using the transperitoneal approach). We extracted mean, standard deviation, and range values in order to obtain the standardized mean difference and risk ratio, 95% CI, and p-value for the overall effect. There are six parameters that we compared; operative time, estimated blood loss, hospital stay, total infection, postoperative ileus, and major complication as shown in S2 Fig, respectively.

There were eight retrospective studies included in this meta-analysis which were published between 2010 and 2022. Three of eight studies are using the laparoscopic approach and five are using the open/classic method. All of the patients in this study underwent radical cystectomy with various types of urinary diversions. In this study, the staging of patients with bladder

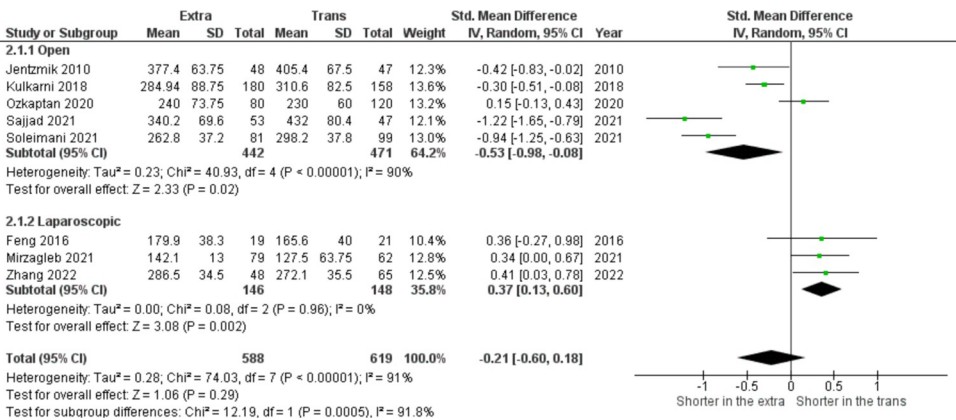

**Fig 1. Operative time (minutes) in extraperitoneal vs transperitoneal approach.**

cancer was quite varied. The follow-up time for each study was also different, ranging from 1 to 70 months overall.

In this meta-analysis of eight studies (n = 1207), Fig 1 shows no significant difference related to operating time between both approaches (SMD = − 0.21, 95% CI: − 0.60 to 0.18, $I^2$ = 91%, z = 01.06, p = 0.29). However, based on the sub-group analysis, operating time favors the extraperitoneal approach for the open-method surgery and is shorter in the transperitoneal approach if it is done laparoscopically.

In Fig 2, among nine studies included (n = 1207), the pooled analysis of estimated blood loss parameters showed that no significant difference was found between both groups (SMD = −0.26, 95% CI: −0.69 to 0.17, $I^2$ = 92%, z = 1.18, p = 0.24). In seven selected studies, (n = 1112), there was heterogeneity among the results ($I^2$ = 96%), therefore the random effect model was used for analysis. The results (Fig 3) showed that pooled analysis of hospital stay duration is not statistically significant (SMD = −0.58, 95% CI: −1.22 to 0.06, $I^2$ = 96%, z = 1.79, p = 0.07).

In the pooled analysis of total infection after the procedure, seven studies (n = 1066) are included. According to the results ($I^2$ = 48%), heterogeneity is acceptable, and therefore the fixed effect model was used for analysis. The results showed that there is no statistically significant difference in total infection events between the two approaches (RR = 0.76, 95% CI: 0.57 to 1.00, $I^2$ = 48%, z = 1.96, p = 0.05) as shown in Fig 4. Another parameter tested for this meta-

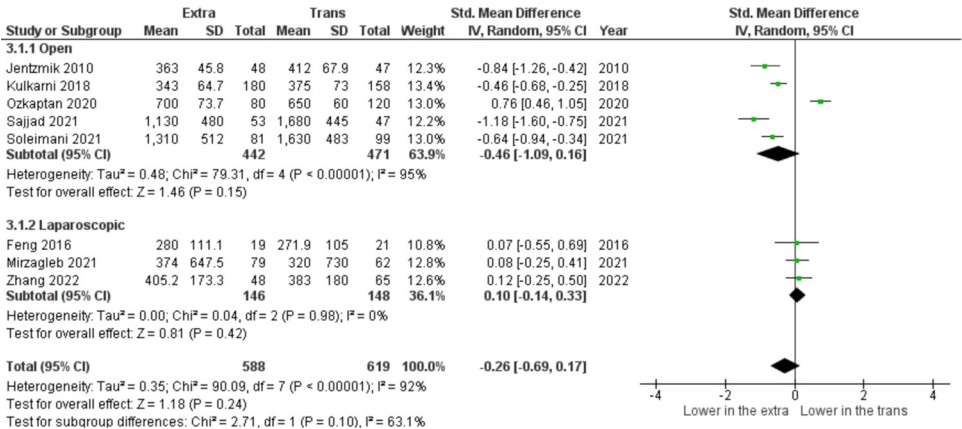

**Fig 2. Estimated blood loss (mL) in extraperitoneal vs transperitoneal approach.**

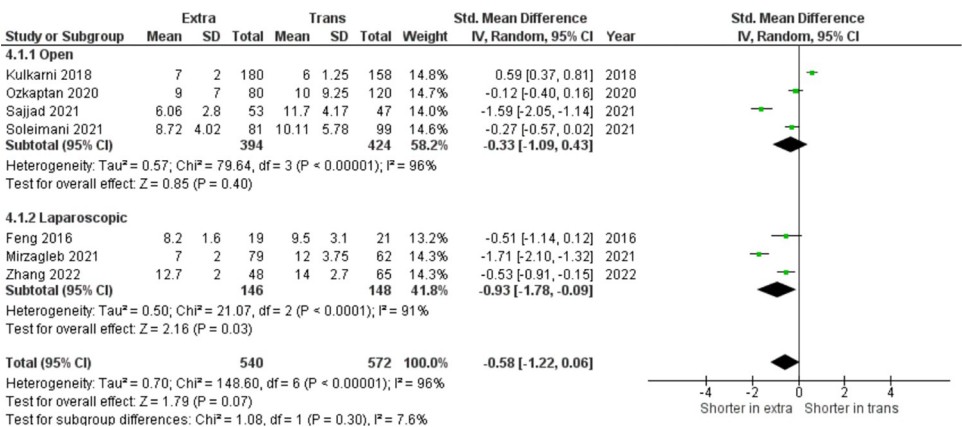

**Fig 3. Hospital stays (days) in extraperitoneal vs transperitoneal approach.**

analysis was shown in Fig 5, including seven (n = 1066), has found that the extraperitoneal cystectomy approach significantly reduced post-operative ileus occurrence compared to the transperitoneal approach (RR = 0.47, 95% CI: 0.37 to 0.59, $I^2$ = 44%, z = 6.19, p <0.00001).

The last outcome parameter included in this meta-analysis is related to major complications after the radical cystectomy according to Clavien-Dindo grade 3–5, which includes five studies (n = 688). According to Fig 6, no statistically significant differences were noted in major complications after surgery between both groups (RR = 0.78, 95% CI: 0.59 to 1.03, $I^2$ = 0%, z = 1.72, p = 0.09). However, when we assessed publication bias using a funnel plot, we obtained asymmetrical results. This may be due to the minimal sample size (S3 Fig).

## Discussion

This study is a systematic review of quantitative data analysis. All eligible retrospective studies were published between 2010 and 2022. To our knowledge, this is the first systematic review and meta-analysis study that compares extraperitoneal and transperitoneal radical cystectomy

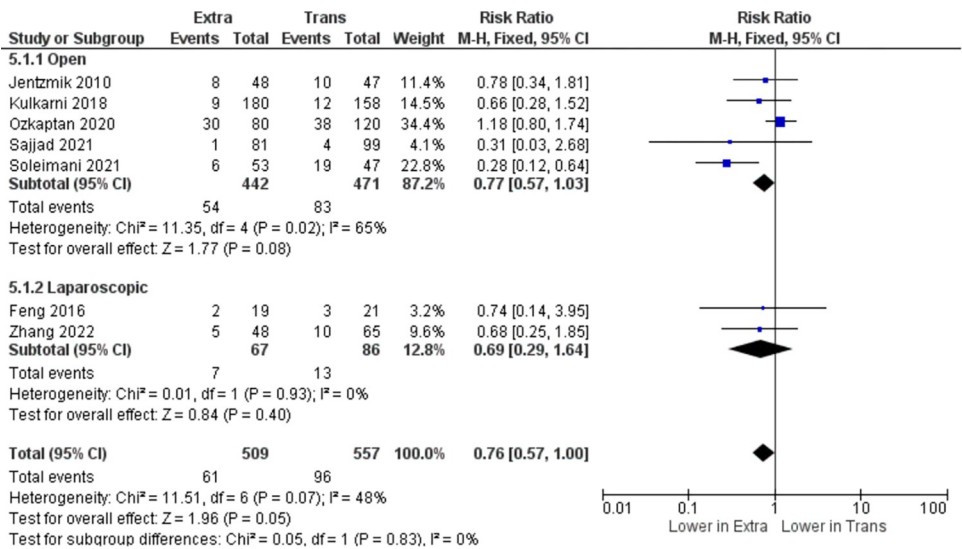

**Fig 4. Total infection (n) in extraperitoneal vs transperitoneal approach.**

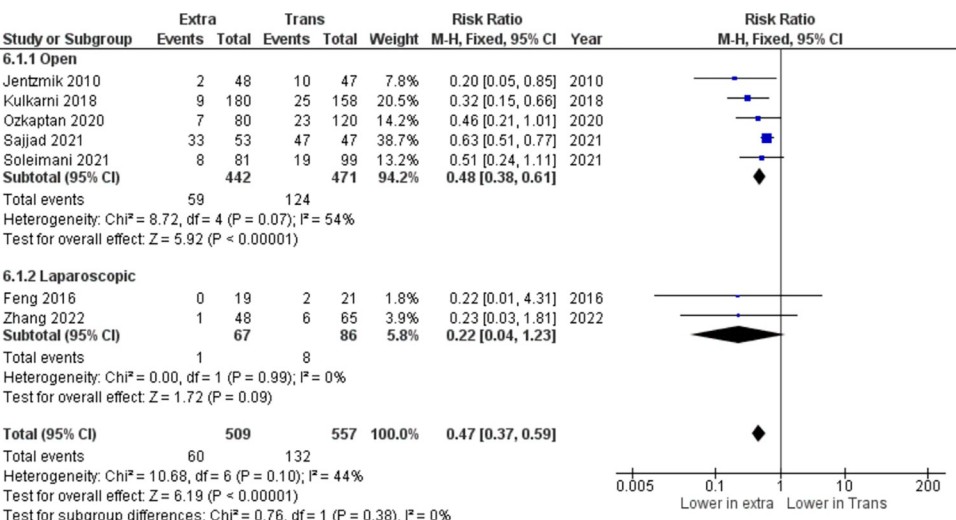

**Fig 5. Post-operative ileus (n) in extraperitoneal vs transperitoneal approach.**

on perioperative and postoperative parameters. For decades, the transperitoneal approach is the standard surgical treatment for non-metastatic muscle-invasive and high-risk non-muscle-invasive bladder cancer [2]. However, complications following the traditional approach have been a growing concern for clinicians. Due to disturbance of the peritoneal membrane, several conditions have emerged such as ileus and infection. The incidence of these complications was as high as 40–44% in a recent study [4]. Kulkarni et al. proposed the extraperitoneal approach to minimize bowel injury and other side effects in 1999 for radical cystectomy, and this technique has been widely performed in many health centers [8]. The aim of the current study was to compare both approaches. The result of our study showed that postoperative ileus and major complication occurrence is significantly lower in the extraperitoneal approach compared to the transperitoneal approach. However, there is no significant correlation between other parameters regarding both approaches.

Difficulty in reducing postoperative gastrointestinal complications, specifically ileus, has been a debated topic regarding transperitoneal radical cystectomy. Ileus consisted of two

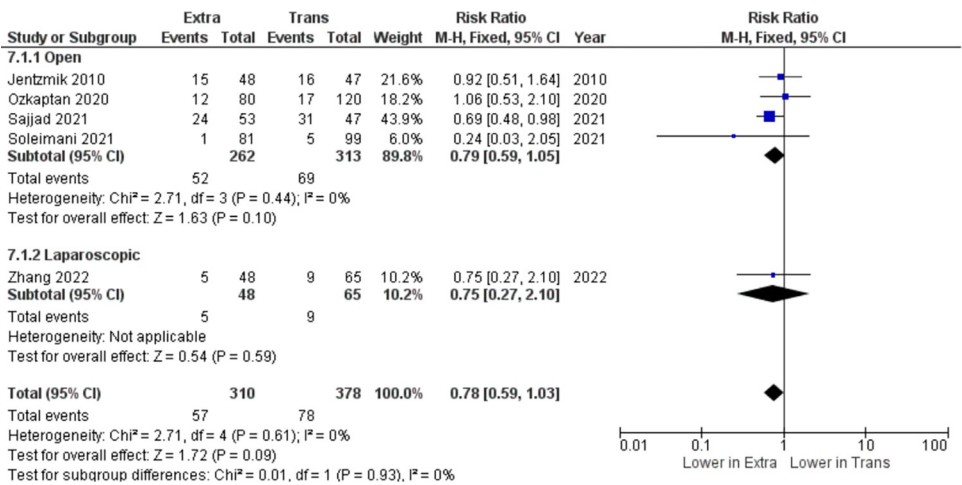

**Fig 6. Major complications (n) in extraperitoneal vs transperitoneal approach.**

types, which are paralytic and obstructive. The mechanism of paralytic ileus remains equivocal. While the mechanism of obstructive ileus largely includes adhesions (accounting for 65%-75% of the cases). Importantly, obstructive ileus is the main cause of reoperation after RC [9]. Many attempts have been made to reduce ileus after RC such as gum chewing, medications targeting the peripheral μ-opioid receptor, and fast-track regimens. Meanwhile, the surgery itself is a much greater risk factor for the occurrence and development of ileus [10]. Keeping the peritoneal continuity, theoretically, is an important milestone in reducing postoperative complications [7]. The first theory is EPRC can localize the peritonitis arising from urine or intestinal leakage. Urinary extravasation and intestinal leakage to the peritoneal cavity from either the anastomotic site or the stump suture are not uncommon during the postoperative period. Therefore, complete retroperitonealization is an effective method to minimize the undesirable outcome of urine or intestinal leakage. Second, although no such leakage complications exist, reconstruction of the peritoneal cavity could cause adhesions and ensuing enteroparalysis or even mechanical obstruction. Finally, avoiding a de-peritonealized pelvic wall will also decrease the risk of a strangulated internal hernia developing after pelvic lymphadenectomy when the patient has tortuous, elongated external iliac arteries [11]. In this meta-analysis, the occurrence of postoperative ileus is significantly lower in patients who underwent RC with extraperitoneal approach (RR = 0.47, 95% CI: 0.37 to 0.59, $I^2$ = 44%, z = 6.19, p <0.00001). Kulkarni, et al. have also agreed that GI complications (ileus p < 0.001 and intestinal obstruction p = 0.002) are significantly lower in EPRC group [8]. Özkaptan, et al. stated that the time for recovery of bowel function, the time for passage of stool, and the rate of postoperative ileus were significantly lower in EPRC group (p < 0.01, p < 0.01 and p < 0.043) respectively) [5]. Soleimani, et al. claimed that early GI complications were lower in EPRC groups, including oral intake intolerance (21 vs. 8, p = 0.04), ileus (19 vs. 8, p = 0.04), intestinal obstruction (3 vs. 0, p = 0.04), and anastomosis leakage (8 vs. 1, p = 0.01) [12]. Several other studies also concluded similar results [13, 14]. In this meta-analysis, we found no significant difference related to major complications occurring after EPRC or TPRC surgery. Along with this finding, none of the studies included in the analysis stated any significant differences between both groups [3, 5, 12–14].

The duration of RC operation and estimated blood loss are not significantly different between EPRC and TPRC groups. We also found that there is great heterogeneity among the journals included in this study on operative time and EBL parameters. The reason for this may be due to differences in laparoscopy or open method used in the operation, differences in surgeon's performance experience, and also different availability of time-saving equipment such as endo-staplers for gut anastomosis, as well as thrombotic products like flowseal, hemolocks, ligaclips, ligasures, etc [15]. Many other authors conclude similarly [5, 13, 16]. However, against the trend, one of the studies associates TPRC with significant reduction in blood loss [9].

We found that the duration of the patient's hospitalization is not significantly different between EPRC and TPRC (p = 0.15). Length of stay is an important measure of the patient's satisfaction and healthcare costs. Many factors may influence the length of stay, which includes the surgical method (open/laparoscopic), younger age, shorter operative time, nasogastric tube removal in the operation theatre, etc. [17]. Because of these multifactorial reasons that could alter the hospitalization time, is not out of the ordinary that high heterogeneity is found ($I^2$ = 96%) between the studies included. However, two authors have claimed that the duration of patient stay is significantly lower EPRC group [3, 14]. Özkaptan et al. and Soleimani et al, also concur that the mean length of hospital stay is shorter in the EPRC group, although not significant [5, 12].

Post-surgery infection is one of the most common complications after abdominal surgery. The type of infections observed in this study includes wound infection, pyelonephritis, urosepsis, pneumonia, urinary tract infection, fever of unknown origin, gastroenteritis, and abdominal/pelvic abscess. TPRC approach destructs the anastomosis of the peritoneal membrane causing a bigger port *de entrée* for bacteria and the spillage of urine or bowel materials inside the peritoneal cavity also leads to infection. However, most studies did not find significant differences in infection rate between both groups [3, 5, 18, 19], the pooled analysis also shows no statistically significant difference in total infection events between the two approaches (RR = 0.76, 95% CI: 0.57 to 1.00, $I^2$ = 48%, z = 1.96, p = 0.05).

There were some limitations in this study. First of all, for a meta-analysis study, we have included a relatively small number of subjects and the radical cystectomy procedures were performed over a wide range of time (>10 years). Presumably, the results may differ in the current development of the medical field. Second, most of the studies included in this meta-analysis are retrospective studies, therefore it is almost impossible to avoid selection bias and attrition bias. Third, the baseline staging and follow-up time are quite varied. Therefore, bias in patient selection and publication bias might be considered. However, the data on readmission rates is lacking. The ERAS protocol was used in one study by Zhang et al. [3]. A recent recommendation has proposed the adoption of the Enhanced Recovery After Surgery (ERAS) protocol as a means to decrease postoperative complications and expedite the recovery process. It is likely responsible for the extended length of their hospital stays in some studies.

## Conclusion

In summary, extraperitoneal radical cystectomy is a safe and feasible surgical strategy for bladder cancer patients. Compared to the transperitoneal approach, the extraperitoneal approach has a significantly lower occurrence of postoperative ileus. However, it is inconclusive whether major complication, estimated blood loss, operative time, hospital stay duration, and total infection are in favor of either approach. Further meta-analysis with larger studies included should be performed.

## Supporting information

**S1 Fig. PRISMA flow diagram.**
(DOCX)

**S2 Fig. Forest plot.**
(DOCX)

**S3 Fig. Funnel plot.**
(DOCX)

**S1 Table. Literature search strategy.**
(DOCX)

**S2 Table. Study characteristics and results.**
(DOCX)

**S3 Table. Risk of bias assessment (Newcastle–Ottawa quality assessment scale criteria).**
(DOCX)

**S4 Table. PICO analysis.**
(DOCX)

## Author Contributions

**Conceptualization:** Nugroho Purnomo, Moammar Andar Roemare Siregar.

**Data curation:** Kevin Leonardo, Nugroho Purnomo, Andika Afriansyah, Moammar Andar Roemare Siregar.

**Formal analysis:** Kevin Leonardo, Andika Afriansyah.

**Investigation:** Kevin Leonardo, Moammar Andar Roemare Siregar.

**Methodology:** Kevin Leonardo, Andika Afriansyah, Moammar Andar Roemare Siregar.

**Supervision:** Hendy Mirza, Doddy Hami Seno, Nugroho Purnomo, Moammar Andar Roemare Siregar.

**Validation:** Moammar Andar Roemare Siregar.

**Writing – original draft:** Kevin Leonardo, Andika Afriansyah, Moammar Andar Roemare Siregar.

**Writing – review & editing:** Hendy Mirza, Doddy Hami Seno.

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
