## [Decision Letter · Decision Letter 0]

22 May 2023

PONE-D-23-05792Transperitoneal vs Extraperitoneal Radical Cystectomy: A systematic review and meta-analysisPLOS ONE

Dear Dr. Siregar,

Thank you for submitting your manuscript to PLOS ONE. After careful consideration, we feel that it has merit but does not fully meet PLOS ONE’s publication criteria as it currently stands. Therefore, we invite you to submit a revised version of the manuscript that addresses the points raised during the review process.

We look forward to receiving your revised manuscript.

Kind regards,

Federico Ferrari, MD, PhD

Academic Editor

PLOS ONE

- https://www.clinical-genitourinary-cancer.com/article/S1558-7673(15)00008-7/fulltext

- file:///D:/C%20DRIVE/Downloads/6147-Main%20Text-27148-3-10-20230104.pdf

- https://ip.ios.semcs.net/articles/bladder-cancer/blc200280

- https://wjso.biomedcentral.com/articles/10.1186/s12957-022-02587-1

In your revision ensure you cite all your sources (including your own works), and quote or rephrase any duplicated text outside the methods section. Further consideration is dependent on these concerns being addressed.

Reviewers' comments:

Reviewer's Responses to Questions

**Comments to the Author**

1. Is the manuscript technically sound, and do the data support the conclusions?

Reviewer #1: Partly

2. Has the statistical analysis been performed appropriately and rigorously? 

Reviewer #1: I Don't Know

3. Have the authors made all data underlying the findings in their manuscript fully available?

Reviewer #1: Yes

4. Is the manuscript presented in an intelligible fashion and written in standard English?

Reviewer #1: Yes

5. Review Comments to the Author

Reviewer #1: This ia a metanalysis on relatively heterogeneous and mostly retrospective/low quality studies on a relevant and interesting topic in treatment of muscle invasive bladder cancer.

This makes the authors' job difficult to draw a conclusion from this types of publications.

My comments are a follows:

1- It is very important to note that the reported complications on Radical cystectomy (RC) is significantly different in retrospective Vs. prospective studies. In honest prospective reports, the complication rates are between 60-80 percent of the patients. One of the paper included in your analysis-reference 9- has reported 0 patients with grade 3 and higher complications. In my view the authors and the publishing journals and the reviewers of this paper should be questioned and if the data is real, all of us should go there and get appropriate training from them. Until then, I do not think that this study should be included in your analysis. I am not sure about the quality of other papers included here as well.

2- Your studies probably matched the baseline criteria here, although I am not sure about it, but your paper does not show this. For example, are the groups have the same rate of Charlson comorbidity index, neoadjuvant chemotherapy rate, similar follow up period, baseline similar staging/variant rate, etc.

3- I do not think you need to review Clavien-Dindo Classification in your discussion. This is well known criteria and you can just cite the reference.

4- I think hospital stay and risk of readmission is probably a better surrogate of all outcome measures here due to heterogeneity. You did not mention the readmission rate.

5- How many of patients in these studies followed ERAS protocol?

6- Your study regardless of my points has more limitation and it is better to be acknowledged by the authors rather than reviewers.

6. PLOS authors have the option to publish the peer review history of their article (what does this mean?). If published, this will include your full peer review and any attached files.

Reviewer #1: **Yes: **Hamidreza Abdi

---

## [Author Response · Author response to Decision Letter 0]

6 Jul 2023

Manuscript title : Transperitoneal vs extraperitoneal radical cystectomy: a systematic review and meta-analysis

> : Author’s response/answer

# Response to reviewer 

1. It is very important to note that the reported complications on Radical cystectomy (RC) is significantly different in retrospective Vs. prospective studies. In honest prospective reports, the complication rates are between 60-80 percent of the patients. One of the paper included in your analysis-reference 9- has reported 0 patients with grade 3 and higher complications. In my view the authors and the publishing journals and the reviewers of this paper should be questioned and if the data is real, all of us should go there and get appropriate training from them. Until then, I do not think that this study should be included in your analysis. I am not sure about the quality of other papers included here as well.

> Thank you for you review for our manuscript. We have replaced the studies we used with only good quality retrospective studies. There are total 8 retrospective studies in this systematic review and meta analysis. We have already re-analyze and obtain the results listed in the revision of this manuscript (S3 Table. Study Characteristics and Results). 

2. Your studies probably matched the baseline criteria here, although I am not sure about it, but your paper does not show this. For example, are the groups have the same rate of Charlson comorbidity index, neoadjuvant chemotherapy rate, similar follow up period, baseline similar staging/variant rate, etc.

> We have some baseline characteristics to the table, such as staging and Charlson score (S3 Table. Study Characteristics and Results). Of the 8 studies we analyzed, the cancer stages in the studies used varied. The follow-up time of each study also differed from 1 to 70 months. Other characteristics were not described in the studies we analyzed.

3. I do not think you need to review Clavien-Dindo Classification in your discussion. This is well known criteria and you can just cite the reference.

> Thank you for your feedback. We have removed the review Clavien-Dindo Classification in our discussion.

4. I think hospital stay and risk of readmission is probably a better surrogate of all outcome measures here due to heterogeneity. You did not mention the readmission rate.

> We've tried to discuss this. However, from what we found from the studies we analyzed, there were no studies that explained the readmission rate.

5. How many of patients in these studies followed ERAS protocol?

> Unfortunately, only 1 of 8 studies assessed follow-up of patients with the ERAS protocol (Zhang et al, 2022). ERAS protocol was not followed in this study which probably accounts for the varied duration of hospital stay.

6. Your study regardless of my points has more limitation and it is better to be acknowledged by the authors rather than reviewers.

> Thank you for the feedback. We have add our limitation of this study because baseline staging and follow up time are quite varied. Therefor, bias in patient selection might be occurred.

---

## [Decision Letter · Decision Letter 1]

29 Aug 2023

PONE-D-23-05792R1Transperitoneal vs Extraperitoneal Radical Cystectomy: A systematic review and meta-analysisPLOS ONE

Dear Dr. Siregar,

Thank you for submitting your manuscript to PLOS ONE. After careful consideration, we feel that it has merit but does not fully meet PLOS ONE’s publication criteria as it currently stands. Therefore, we invite you to submit a revised version of the manuscript that addresses the points raised during the review process.

We look forward to receiving your revised manuscript.

Kind regards,

Atalel Fentahun Awedew, MD,MPH

Academic Editor

PLOS ONE

Journal Requirements:

Additional Editor Comments:

1. No introduction in the abstract

2. Submit PRISMA checklist as supplement file

3. Which meta-analysis model?( Mantel-Haenszel model…..

4. Why RR for report analysis? It is better used OR for cross sectional stud

5. Method of heterogeneity measurement and techniques of heterogeneity handling (sensitivity, subgroup analysis and meta regression)

6. The research question (objectives) should be clearly addressed in accordance with PICO approach

7. Publication biases ?

Reviewers' comments:

Reviewer's Responses to Questions

**Comments to the Author**

1. If the authors have adequately addressed your comments raised in a previous round of review and you feel that this manuscript is now acceptable for publication, you may indicate that here to bypass the “Comments to the Author” section, enter your conflict of interest statement in the “Confidential to Editor” section, and submit your "Accept" recommendation.

Reviewer #1: All comments have been addressed

2. Is the manuscript technically sound, and do the data support the conclusions?

Reviewer #1: Yes

3. Has the statistical analysis been performed appropriately and rigorously? 

Reviewer #1: I Don't Know

4. Have the authors made all data underlying the findings in their manuscript fully available?

Reviewer #1: Yes

5. Is the manuscript presented in an intelligible fashion and written in standard English?

Reviewer #1: Yes

6. Review Comments to the Author

Reviewer #1: Thanks for addressing my points.

I like the way it is written now. I leave the following issues up to you if you want to address then in your discussion:

The data on readmission rate is lacking.

The ERAS protocol was used in one study, and the finding of that particular study is in line with the results of meta-analysis.

7. PLOS authors have the option to publish the peer review history of their article (what does this mean?). If published, this will include your full peer review and any attached files.

Reviewer #1: No

---

## [Author Response · Author response to Decision Letter 1]

5 Nov 2023

Manuscript PONE-D-23-05792

Author’s response / answer

Reviewer #1: Thanks for addressing my points.

I like the way it is written now. I leave the following issues up to you if you want to address then in your discussion:

The data on readmission rate is lacking.

The ERAS protocol was used in one study, and the finding of that particular study is in line with the results of meta-analysis.

We have responded to your comments below,

# Response to reviewer 1

Thank you for your constructive comments and suggestions for our manuscript, we are very pleased with your comments. We have revised the manuscript, we also have checked the comments, and revised the manuscript accordingly. We hope the revised version adds more value and is more suitable for publication.

Additional Editor Comments:

1. No introduction in the abstract

Thank you for your comment. We have already changed the subtitle from Objectives to Background. The introduction of the manuscript is mentioned on that section

2. Submit PRISMA checklist as a supplement file

Yes, we have added PRISMA checklist and PRISMA Abstract checklist as supplement file

3. Which meta-analysis model? 

The type of meta-analysis model that we used was The Cochran-Mantel-Haenszel Method

4. Why RR for report analysis? It is better used OR for cross sectional study

Thank you for your comment. We chose Relative Risk due to the study design (cohort) that was included in this study. 

5. Method of heterogeneity measurement and techniques of heterogeneity handling (sensitivity, subgroup analysis and meta regression)

I2 was determined using RevMan version 5.4 in order to look into statistical heterogeneity

6. The research question (objectives) should be clearly addressed in accordance with PICO approach

Thank you for your kind advice. We have mentioned our main objective is to compare the transperitoneal RC approach with the extraperitoneal RC approach in the treatment of bladder cancer patients based on some outcomes that can be seen as efficacy and safety profiles of both approaches

7. Publication biases?

We have added funnel plots in the supplemental file as our way to check the publication biases

1. Please ensure that you refer to Figure 1 in your text as, if accepted, production will need this reference to link the reader to the figure.

2. Please upload a copy of Figure 1. Or if the figure is no longer to be included as part of the submission please remove all references to it within the text.

# Response to Additional confirmation

Thank you for your comments related to the missing Figure 1 issue. We are really sorry to inform you that Figure 1 has been changed to S2_Figure in the Supplemental files folder (PRISMA Flow Diagram). So, it is no longer to be included as part of the manuscript. We would like to erase and remove all the references related to it within the text. Figure 2-7 in the text is modified to Figure 1-6 as the impact of the change itself. Thank you for your kind understanding

---

## [Editor Report · Decision Letter 2]

10 Nov 2023

Transperitoneal vs Extraperitoneal Radical Cystectomy: A systematic review and meta-analysis

PONE-D-23-05792R2

Dear Dr. Kevin Leonardo

We’re pleased to inform you that your manuscript has been judged scientifically suitable for publication and will be formally accepted for publication once it meets all outstanding technical requirements.

Kind regards,

Atalel Fentahun Awedew, MD,MPH

Academic Editor

PLOS ONE
---

## [Editor Report · Acceptance letter]

20 Nov 2023

PONE-D-23-05792R2 

Transperitoneal vs extraperitoneal radical cystectomy: a systematic review and meta-analysis 

Dear Dr. Siregar:

I'm pleased to inform you that your manuscript has been deemed suitable for publication in PLOS ONE. Congratulations! Your manuscript is now with our production department. 

Kind regards, 

on behalf of

Dr. Atalel Fentahun Awedew 

Academic Editor

PLOS ONE